# REDUCE, REUSE, AND RECYCLE: NAVIGATING TEST-TIME ADAPTATION WITH OOD-CONTAMINATED STREAMS

## ABSTRACT

Test-Time Adaptation (TTA) aims to quickly adapt a pre-trained Deep Neural Network (DNN) to shifted test data from unseen distributions. Early TTA works only targeted simple and restrictive test scenarios that did not align with the philosophy of TTA that emphasizes practicality. Subsequent research efforts have thus been geared towards exploring more realistic test scenarios. In the same spirit, this work investigates for the first time TTA with data streams contaminated with out-of-distribution (OOD) data. Surprisingly, we observe the existence of benign OOD data that can improve TTA performance. We provide meaningful insights into the causes of benign OOD-contamination by analyzing the feature space of the pre-trained DNN. Inspired by these empirical findings, we propose R3, a novel TTA algorithm that specifically targets OOD-contaminated streams. Our experimental results verify that R3 improves competitive baselines by up to nearly 3%p on OOD-contaminated streams created with CIFAR-10-C and ImageNet-C.

## 1 INTRODUCTION

Powered by enormous datasets and computational resources, Deep Neural Networks (DNNs) continue to push the boundaries of machine intelligence (LeCun et al., 2015). DNNs, however, are still far from being the omnipotent machine learning model they are mistakenly advertised to be. One key limitation of DNNs is their failure to generalize to corrupted or shifted test data (Pan & Yang, 2010), which is referred to as shifted in-distribution (InD) data from here on. Improving their robustness to diverse distribution shifts thus remains a critical challenge when deploying DNNs in an open world.

Transductive inference aims to address the aforementioned limitation by adapting DNNs to specific distribution shifts (Gammerman et al., 2013). Because it is infeasible to anticipate the myriad of distribution shifts that may occur, transductive inference offers a more realistic solution than its inductive counterpart. Test-Time Adaptation (TTA) (Wang et al., 2020) distinguishes itself from other methods in transduction, such as Unsupervised Domain Adaptation (UDA) (Ganin & Lempitsky, 2015) and Test-Time Training (TTT) (Sun et al., 2020), in two distinct manners: 1) no direct access to training data is allowed, and 2) test data arrive in an online manner and cannot be re-visited. These assumptions make TTA the most practical approach to transduction.

Unlike early works in TTA that dealt with relatively simple and mild test scenarios (Niu et al., 2023), more recent research efforts are being directed towards exploring more challenging yet realistic test scenarios for TTA. For instance, Gong *et al.* (Gong et al., 2022) study temporally correlated, instead of i.i.d. data streams, and Niu *et al.* (Niu et al., 2023) investigate TTA in a "dynamic wild world," where test data have mixed and class-imbalanced distribution shifts. In the same vein, this work pioneers TTA with data streams that are contaminated with irrelevant out-of-distribution (OOD) data. As depicted in Figure 1, once deployed in an open world, the pre-trained DNN will inevitably encounter OOD data. Hence, investigation of the proposed test scenario, dubbed *OOD-contaminated streams*, allows for safer and more flexible deployment of pre-trained DNNs to a wide range of test scenarios without constraints.

Contrary to the popular belief that OOD data undermine the reliability of DNNs (Yang et al., 2021), we observe that performing TTA with OOD-contaminated streams improves the adaptation performance on some shifted InD data. This unusual observation alludes to the existence of "benign

Figure 1: (a) Prior to deployment, the DNN is pre-trained on clean InD data, *e.g.,* data collected on a sunny day. (b) In the real world, the deployed DNN encounters a rainy condition and must be adapted accordingly. Unfortunately, current TTA protocols cannot handle OOD data that exist outside the label set of the pre-training data. (c) The conceptual overview of R3: ① Harmful test instances are reduced with two filtering thresholds. ② Remaining instances are reused via data mixup and contrastive learning. ③ Filtered instances are recycled into an auxiliary loss.

OOD-contamination," whose unrealized potential opens new doors for performance improvement that are unique to OOD-contaminated streams. To understand the cause of benign OOD-contamination, we analyze the feature space of the pre-trained DNN. Our analysis reveals that the feature spaces resided by shifted InD and OOD data overlap significantly, and as a result, OOD data that share the feature space with shifted InD data have the ability to facilitate the domain transfer from clean to shifted InD data. The highly entangled nature of the two data results in the additional side-effect of discarding shifted InD data when filtering out OOD data.

Inspired by these findings, we propose R3 (Reduce, Reuse, and Recycle), a novel TTA algorithm designed specifically for OOD-contaminated streams. To minimize the loss of shifted InD data, R3 conservatively identifies and *reduces* the amount of OOD-contamination with two cost-efficient metrics. The detected OOD instances are *recycled* into an auxiliary loss that drives their predictions closer to the Uniform distribution, effectively preventing the transfer of undesirable features. The remaining instances are *reused* for similarity-based mixup and contrastive learning with class-wise prototypes; these additional signals allow the pre-trained DNN to more robustly fit shifted InD data while preserving the original feature space. We demonstrate that R3 achieves state-of-the-art performance on two benchmark datasets, CIFAR-10-C and ImageNet-C, contaminated with various types of OOD data. Our contributions are largely three-fold:

- This is the first work to explore test-time model adaptation with OOD-contaminated streams that contain both shifted InD and irrelevant OOD data. Because it is impossible for the deployed DNN to evade OOD data in an open world, the proposed test scenario is far more realistic than TTA on curated data streams that explicitly contain targeted distribution shifts.

- We reveal the existence of beneficial OOD data that can assist in improving the adaptation performance on shifted InD data. Further examination of the pre-trained DNN's feature space brings to light that this surprising phenomenon is a result of the highly entangled nature of shifted InD and OOD data.

- We propose R3, a novel TTA algorithm that targets OOD-contaminated streams. Our extensive experimental results on diverse combinations of shifted InD and OOD data demonstrate the superiority of R3 to strong baselines from the TTA literature.

## 2 RELATED WORKS

### 2.1 TEST-TIME ADAPTATION AND ITS PROGRESSION

Test-Time Adaptation (TTA) is a popular branch of transductive inference (Gammerman et al., 2013) that concerns with adapting a pre-trained DNN to specific distribution shifts. The particular appeal of TTA, compared to other variants in transductive inference (*e.g.,* UDA (Ganin & Lempitsky, 2015; Ganin et al., 2016) and SFDA (Li et al., 2020b; Ding et al., 2022; Lee et al., 2022)), lies in its practicality. TTA performs adaptation without direct access to train data and under an online setting, in which test data cannot be revisited. To account for the lack of labels in test data, many of TTA methods perform adaptation by minimizing the entropy of the pre-trained DNN's softmax predictions on test data (Wang et al., 2020). Moreover, fully TTA methods (Boudiaf et al., 2022; Lim et al., 2023; Jang & Chung, 2022; Zhang et al., 2022; Choi et al., 2022) only update affine parameters and statistics of Batch Normalization layers (Ioffe & Szegedy, 2015) for fast and cost-efficient adaptation.

In the beginning, TTA only targeted elementary test scenarios, in which test data could be gathered into a batch to perform batch-wise adaptation, and each individual batch was sampled from the same shifted distribution. More recent works are starting to reflect additional obstacles that may arise in an open world. Continual TTA (Wang et al., 2022b; Song et al., 2023; Döbler et al., 2022) studies TTA in a non-stationary and continually changing test environment. Several works (Zhao et al., 2023; Gong et al., 2022) tackle class-imbalanced or temporally dependent data streams. Single-image TTA (Khurana et al., 2021) explores an extreme setting where a single test instance arrives at a time. Niu *et al.* (Niu et al., 2023) consolidate the above scenarios into a single setting named TTA in a dynamic wild world. To stably perform adaptation in more challenging scenarios, recent methods allow usage of auxiliary signals, such as data augmentation (Khurana et al., 2021) and/or partial information about source data (Döbler et al., 2022; Niu et al., 2022) even at the cost of increased computation and memory consumption. Our work is closely related to Open-set or Universal Domain Adaptation (Panareda Busto & Gall, 2017; Saito et al., 2018; You et al., 2019; Saito et al., 2020), but to the best of our knowledge, this is the first work to study open-world data streams in TTA.

## 2.2 DETECTION AND UTILIZATION OF OUT-OF-DISTRIBUTION DATA

The presence of OOD data is often believed to significantly degrade the performance and reliability of DNNs. This conventional belief in machine learning has inspired research efforts to successfully detect and exclude OOD data from the inference process (Yang et al., 2021). The most straightforward approach to OOD detection is to utilize the DNN's predictive confidence scores (Hendrycks & Gimpel; Lee et al., a). Unfortunately, DNNs often output miscalibrated and over-confident predictions on OOD data, making confidence scores an unreliable indicator of OOD-ness (Guo et al., 2017). To overcome the drawback of confidence scores, a plethora of new OOD detection metrics, based on the energy-based interpretation of DNNs (Liu et al., 2020), temperature scaling (Liang et al.), rectified activations (Sun et al., 2021), virtual logit matching (Wang et al., 2022a), and various distance measures (Lee et al., 2018) have been suggested. The influence of OOD data is actively studied in various fields of research, including but not limited to: continual learning (Bang et al., 2022) and semi-supervised learning (Huang et al., 2021). Against the long-held belief that OOD data are harmful, some recent studies revealed that OOD data, when leveraged appropriately, can improve the generalization performance of DNNs (Park et al., 2021; Lee et al., b; Wei et al., 2022; Bai et al.).

## 3 MOTIVATION

### 3.1 EXISTENCE OF BENIGN OOD-CONTAMINATION

It is easy to assume that performing TTA on shifted InD and OOD data together will lead to sub-optimal performance due to the distributional mismatch between the two. Interestingly enough, we reveal that the presence of OOD data in test streams does not always deteriorate the adaptation performance; we dub the subset of OOD data that can assist, rather than harm, the adaptation performance "benign OOD-contamination." In Figure 2, we compare the adaptation perfor-

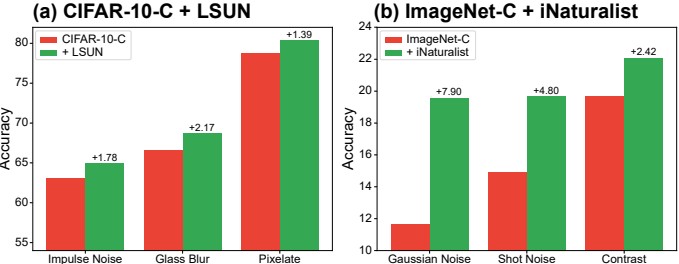

Figure 2: Comparison of TENT adaptation performance with and without OOD-contamination on (a) CIFAR-10-C and (b) ImageNet-C streams. LSUN and iNaturalist are used as OOD-contamination for CIFAR-10-C and ImageNet-C, respectively.

mance of vanilla TENT (Wang et al., 2020), the most widely-adopted TTA approach, on test streams with and without OOD-contamination. We assume that shifted InD and OOD data lie in the same shifted domain. Figure 2 (a) visualizes the results on three different types of CIFAR-10-C streams after inducing OOD-contamination with LSUN (Liang et al., 2017). Surprisingly, executing TENT with LSUN yields higher classification accuracy on CIFAR-10-C. In Figure 2 (b), the same trend is once again observed for ImageNet-C streams before and after OOD-contamination with iNaturalist (Van Horn et al., 2018), demonstrating that the existence of benign OOD-contamination is

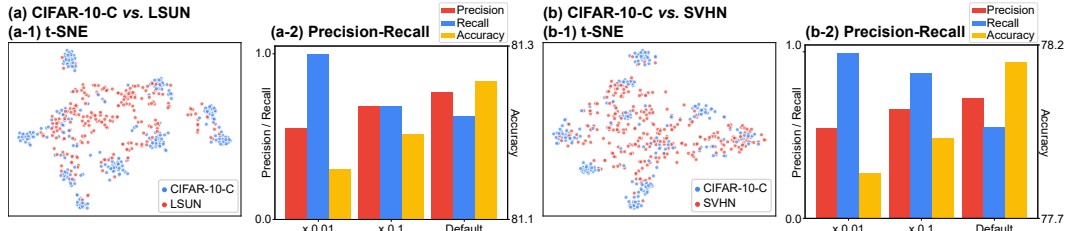

Figure 3: T-SNE and precision-recall analyses of CIFAR-10-C (a) + LSUN and (b) + SVHN. Shifted InD and OOD data are generated from the "Defocus Blur" domain. T-SNE plots show that in the feature space of the pre-trained DNN, shifted InD and OOD data appear nearly inseparable from each other. In precision-recall plots, precision and recall quantify the OOD detection performance on LSUN/SVHN, and accuracy indicates the adaptation performance on CIFAR-10-C. The adaptation accuracy (yellow) exhibits an inverse correlation with recall (blue) in OOD-contaminated streams.

not an isolated incident on smaller datasets. This empirical observation provides a tangible piece of evidence for benign OOD-contamination. In the next section, we analyze the feature space of the pre-trained DNN to offer insights into the phenomenon of benign OOD-contamination.

## 3.2 EMPIRICAL ANALYSIS OF THE FEATURE SPACE

We employ t-SNE (Van der Maaten & Hinton, 2008) to visualize features extracted from penultimate layers of pre-trained DNNs. Figure 3 (a-1) and (b-1) show t-SNE of features of two OOD-contaminated streams: CIFAR-10-C + LSUN and + SVHN. In t-SNE plots, features of CIFAR-10-C and OOD data are colored in blue and red, respectively. Visualization results demonstrate that shifted InD and OOD data appear entangled in the feature space of the pre-trained DNN. This relationship is preserved even in the CIFAR-10-C + SVHN stream even though SVHN is commonly considered to be "far OOD" data that are relatively easier to detect. Such an empirical observation implies that shifted InD and OOD data share domain-specific characteristics, *e.g.,* weather or lighting condition, of the shifted domain. Because Batch Norm layers largely consist of domain-specific information (Li et al., 2017; Schneider et al., 2020), the adaptation protocol of TTA that only updates Batch Norm parameters can be interpreted as performing domain transfer from clean to shifted InD data while preserving the domain-invariant features of InD data. Consequently, during TTA, OOD data that belong in the same domain can provide auxiliary signals about the new shifted domain, thereby contributing to the adaptation performance (Chang et al., 2019; Kang et al., 2019).

A non-negligible side effect of this highly-entangled feature space is that undesirable loss of informative signals inevitably occurs during the filtering process. To corroborate this claim, we perform TENT with three different values of filtering thresholds based on the predictive entropy of a pre-trained DNN; test instances with higher predictive entropy than the threshold are excluded from the adaptation process. The default threshold is set to be $\log(1000) * 0.04$. In Figure 3 (a-2) and (b-2), we show changes in TENT performance according to different threshold values. The left axis shows the OOD detection performance in precision and recall, and the right axis shows the classification accuracy on InD data. TENT performs best in the high precision-low recall region of OOD detection, where OOD data are filtered in a conservative manner, and its performance starts to deteriorate as the precision decreases. Therefore, we deduce that preserving InD data is equally as important as removing harmful OOD-contamination, necessitating a more rigorous form of OOD filtering.

## 4 METHODOLOGY

In this section, we introduce R3, our proposed approach to TTA with OOD-contaminated streams. R3 first reduces the amount of wasteful data by conservatively identifying harmful OOD instances with two cost-efficient metrics for OOD-ness. The identified OOD instances are later recycled and reformulated into an auxiliary loss function that is designed to facilitate a selective transfer of features (Section 4.1). Afterwards, R3 implements similarity-based mixup and contrastive learning with class-wise prototypes by reusing the unfiltered instances (Section 4.2).

**Preliminaries and Notations:** Let us assume that we have a DNN $f_\theta(x)$ that is parameterized with learnable parameters $\theta$ and has been pre-trained on clean InD data $\mathcal{D}_{\mathrm{tr}} = \{(x_{\mathrm{tr}}^i, y_{\mathrm{tr}}^i)\}_{i=1}^{L_{\mathrm{tr}}}$,

where $x_{tr}^i \in \mathcal{X}_{tr}$ and $y_{tr}^i \in \mathcal{C}_{tr}$. TTA aims to adapt $f_\theta(x)$ on arbitrary test data $\mathcal{D}_{te} = \{(x_{te}^i)\}_{i=1}^{L_{te}}$. Unless specified otherwise, TTA mitigates the absence of test labels by employing the entropy minimization loss (Wang et al., 2020): $\min - \sum_c y'_{te,c} \log(y'_{te,c})$, where $y'_{te,c} = f_\theta(c|x_{te})$ denotes the DNN's predictions on a class $c$. Features extracted from the penultimate layer of the pre-trained DNN are denoted by $z \in \mathcal{Z}$, where $\mathcal{Z}$ corresponds to the feature space. Then, the per-class mean of features computed on the clean train data, *i.e.,* class-wise prototypes, can be denoted as: $r_c = (\sum \mathbb{1}[y_{tr}^i = c] z_{tr}^i)/\sum \mathbb{1}[y_{tr}^i = c]$.

In this work, we consider a challenging test scenario in which $\mathcal{D}_{te}$ consists of both shifted but label-sharing InD data of our interest ($\tilde{\mathcal{D}}_{te} = \{(\tilde{x}_{te}^i)\}_{i=1}^{\tilde{L}_{te}}$) and irrelevant OOD data ($\hat{\mathcal{D}}_{te} = \{(\hat{x}_{te}^i)\}_{i=1}^{\hat{L}_{te}}$): by definition, $\tilde{\mathcal{D}}_{te} \cup \hat{\mathcal{D}}_{te} = \mathcal{D}_{te}$, and $\tilde{\mathcal{D}}_{te} \cap \hat{\mathcal{D}}_{te} = \emptyset$. $\tilde{\mathcal{D}}_{te}$ shares the same label set as train data but has different data distribution: $\tilde{\mathcal{X}}_{test} \neq \mathcal{X}_{train}, \tilde{\mathcal{C}}_{test} = \mathcal{C}_{train}$. In the case of $\hat{\mathcal{D}}_{te}$, however, its data and label sets both diverge away from those of $\mathcal{D}_{train}$: $\hat{\mathcal{X}}_{test} \neq \mathcal{X}_{train}, \hat{\mathcal{C}}_{test} \neq \mathcal{C}_{train}$.

## 4.1 REDUCE & RECYCLE: CONSERVATIVE OOD DATA FILTERING

R3 employs two metrics to identify harmful OOD instances prior to performing the adaptation process. For the sake of conciseness, notations for a test instance $x_{te}$ and the associated DNN output $y'_{te}$ is simplified as $x$ and $y'$ from here on. The first measure of OOD-ness in R3 is the energy score (Liu et al., 2020), defined to be: $ES(x; f_\theta) = -T \cdot \log \sum_i^{\mathcal{C}} e^{y'_i/T}$, where $T$ is equivalent to the temperature of the Softmax function. The energy score modifies the naïve confidence score to reduce over-confident predictions on OOD data and increase the separability between InD and OOD data.

However, unlike clean InD data, shifted InD appear intricately entangled with OOD data in the feature space of a pre-trained DNN as shown in Section 3.2. Consequently, the energy score alone remains an insufficient measure for identifying harmful OOD data without sacrificing shifted InD data that we wish to preserve. R3 thus introduces a second measure based on the cosine similarity between the features and class-wise prototypes:

$$CS(x; f_\theta) = \max_c \frac{r_c \cdot z}{||r_c|| \, ||z||}, \quad \text{for } c \in \mathcal{C}_{tr}. \tag{1}$$

$r_c$ is pre-computed on clean InD data following the pre-training stage and is not updated during the adaptation stage. $CS$ essentially quantifies the similarity between the penultimate feature of a test instance and the closest class-wise prototype. With these two metrics at hand, R3 filters out a test instance only if its negative energy score and cosine similarity both fall below pre-set thresholds:

$$\text{If } \mathbb{1}[-ES(x) < \tau_{es}] \cdot \mathbb{1}[CS(x) < \tau_{cs}] = \begin{cases} 1, & \text{then,} \quad x^i \in S_M \\ 0, & \text{then,} \quad x^i \in S_B. \end{cases} \tag{2}$$

$\tau_{es}$ and $\tau_{cs}$ are filtering thresholds for $ES$ and $CS$, respectively, and are treated as separate hyper-parameters of R3. By introducing a more rigorous set of criteria for eliminating test instances, R3 effectively minimizes the chances of shifted InD instances being unintentionally excluded from the adaptation process. $S_B$ and $S_M$ refer to a set of InD and OOD instances, identified from the DNN's viewpoint, and an instance in $S_B$ and $S_M$ is denoted by $x_B$ and $x_M$.

Instead of discarding $S_M$, R3 recycles them into an auxiliary loss that minimizes the cross entropy between the Uniform distribution and the DNN's predictions on instances in $S_M$: $\mathcal{L}_{Unif} = \min - \sum_c \frac{1}{c} \log(y'_{M,c})$. This loss function has been shown to improve the DNN's generalization performance by preventing it from learning irrelevant features (Lee et al., b). Consequently, when utilized for TTA, it can enforce a selective transfer of features that are useful for the shifted domain.

## 4.2 REUSE: SIMILARITY-BASED MIXUP & CONTRASTIVE LEARNING WITH CLASS-WISE PROTOTYPES

The absence of test labels creates an inherently noisy learning signal, which is exacerbated by the conservative OOD filtering scheme in R3. To robustify the adaptation process, we employ mixup (Zhang et al., 2017), a popular form of noise-robust learning (Berthelot et al., 2019; Li et al., 2020a). Original mixup randomly mixes two images in a batch with a mixup coefficient sampled from a beta distribution. R3 modifies the data mixing process by adopting the maximum cosine

similarity between the feature and class-wise prototypes as a novel mixup coefficient:

$$x_{\text{mix}} = \alpha \cdot x_{\text{B}}^i + (1 - \alpha) \cdot x_{\text{B}}^j, \quad \text{where} \ \alpha = CS(x_{\text{B}}^i; f_\theta). \qquad (3)$$

Because $CS(x_{\text{B}}^i; f_\theta)$ is readily available from the previous step, R3's interpretation of mixup does not incur any additional cost compared to the original mixup. With the redefined mixup coefficients, the instances that are closer to class-wise prototypes are weighted more heavily, whereas the opposite holds true for the instances that are farther away from class-wise prototypes. After performing similarity-based mixup, we repeat the filtering process in Eq. (2) to select $S_{\text{mix}}$. Unlike the original mixup, R3 cannot perform mixup in the label space due to the lack of test labels. Instead, R3 defines a separate entropy minimization loss with the instances in $S_{\text{mix}}$: $\mathcal{L}_{\text{mix}} = \min - \sum_c y'_{\text{mix},c} \log (y'_{\text{mix},c})$.

Lastly, R3 performs supervised contrastive learning (Khosla et al., 2020) between penultimate features $z$ of $S_{\text{B}}$ and $S_{\text{mix}}$ and the class-wise prototypes:

$$\mathcal{L}_{\text{Cont}} = - \sum_i \frac{\exp(\text{sim}(z_i, r_c)) \ / \ T_{\text{con}}}{\sum_{c' \in \mathcal{C}_{\text{tr}}} \exp(\text{sim}(z_i, r_{c'})) \ / \ T_{\text{con}}}, \quad \text{where} \ T_{\text{con}} = 0.1. \qquad (4)$$

$\mathcal{L}_{\text{Cont}}$ prevents the feature space from deviating much from that of clean InD data during the adaptation process. Because R3 only updates Batch Norm parameters, it is preferable to retain the feature space of the penultimate layer while alleviating the domain shift, such that the classification layer, which is frozen after the pre-training stage, can map features to correct classes at ease.

**Overall Optimization Scheme:** The final learning objective of R3 can be expressed as: $\mathcal{L}_{\text{R3}} = \lambda_{\text{Ent}} \, \mathcal{L}_{\text{Ent}} + \lambda_{\text{Mix}} \, \mathcal{L}_{\text{Mix}} + \lambda_{\text{Cont}} \, \mathcal{L}_{\text{Cont}} + \lambda_{\text{Unif}} \, \mathcal{L}_{\text{Unif}}$, where $\lambda$ is the coefficient for the associated loss term and is treated as a hyperparameter. The pseudo-code for R3 that encompasses all of the above components and information on how to tune relevant hyperparameters prior to deployment are included in Sections A6 and A7 of Appendix.

## 5 EXPERIMENTS

### 5.1 EXPERIMENTAL SET-UP

**Baselines:** Detailed description of baseline TTA approaches used for comparison can be found in Section A1 of Appendix. These baselines, carefully selected from TTA literature, are a fair reflection of the state-of-the-art in TTA research.

**Datasets and Implementation Details:** R3 and compared approaches are verified on two types of shifted InD data: CIFAR-10-C (Krizhevsky et al., 2009) and ImageNet-C (Deng et al., 2009). When adapting the DNN on CIFAR-10-C, we consider the following types of OOD-contamination: LSUN (Crop) (Liang et al., 2017), SVHN (Netzer et al., 2011), and Describable Textures Dataset (DTD) (Cimpoi et al., 2014). In the case of ImageNet-C, we induce OOD-contamination with iNaturalist (Van Horn et al., 2018) and DTD. Realistically, OOD data would appear under the same domain or corruption as shifted InD data. Therefore, we apply the same set of corruptions to OOD data following the protocol provided by Hendrycks *et al.* (Hendrycks & Dietterich, 2018). Due to the page constraint, implementation details, including the choice of architectures, pre-training protocols, and test stream configurations, can be found in Section A1 of Appendix.

### 5.2 VERIFICATION UNDER SEPARATE AND MIXED CORRUPTION SCENARIOS

We first consider the case where each type of shifted InD data arrives in a separate manner with clear boundaries. We assume that the OOD data are corrupted in the same manner as InD data. The results in terms of the average classification accuracy across all fifteen corruptions types are reported in Table 1. The table also includes standard deviations of adaptation performances computed over five different random seeds. To provide an empirical upper bound performance for reference, we report the performance of TENT on a clean, uncontaminated stream that only contains shifted InD data ("Clean Stream"). On the LSUN-contaminated stream, R3 exhibits further performance improvement from TENT. Furthermore, R3 consistently achieves the best performance on streams contaminated with SVHN and DTD. Note that out of the three OOD datasets, only LSUN, which is "near OOD" (Sastry & Oore, 2020), improves the adaptation performance of TENT. This result supports that near OOD data that bear close resemblance to shifted InD data are more likely to benefit the adaptation process.

Table 1: Comparison against state-of-the-art TTA methods under the **separate** corruption scenario. We report the classification accuracy (%) on CIFAR-10-C averaged over all 15 corruption types. The best result under each OOD-contaminated stream is marked in bold.

| Stream | Method | LSUN | SVHN | DTD | LSUN + SVHN | LSUN + DTD | DTD + SVHN |
|---|---|---|---|---|---|---|---|
| Separate | Test | | | | 63.31 | | |
| | BN | $76.17_{\pm 0.15}$ | $69.74_{\pm 0.62}$ | $68.54_{\pm 0.13}$ | $72.69_{\pm 0.20}$ | $73.01_{\pm 0.05}$ | $68.92_{\pm 0.09}$ |
| | TENT | $76.39_{\pm 0.17}$ | $69.74_{\pm 0.34}$ | $69.48_{\pm 0.20}$ | $73.39_{\pm 0.10}$ | $73.44_{\pm 0.08}$ | $69.61_{\pm 0.07}$ |
| | TENT$_f$ | $76.17_{\pm 0.15}$ | $67.57_{\pm 0.56}$ | $69.31_{\pm 0.32}$ | $71.35_{\pm 0.72}$ | $73.39_{\pm 0.17}$ | $68.23_{\pm 0.30}$ |
| | EATA | $76.12_{\pm 0.12}$ | $69.81_{\pm 0.56}$ | $68.25_{\pm 0.17}$ | $74.09_{\pm 0.54}$ | $73.05_{\pm 0.13}$ | $68.52_{\pm 0.16}$ |
| | SAR | $76.17_{\pm 0.14}$ | $69.75_{\pm 0.63}$ | $68.53_{\pm 0.13}$ | $72.69_{\pm 0.15}$ | $73.01_{\pm 0.07}$ | $68.92_{\pm 0.09}$ |
| | R3 | $\mathbf{76.64}_{\pm 0.13}$ | $\mathbf{72.08}_{\pm 0.20}$ | $\mathbf{72.41}_{\pm 0.27}$ | $\mathbf{75.16}_{\pm 0.15}$ | $\mathbf{75.10}_{\pm 0.30}$ | $\mathbf{71.99}_{\pm 0.24}$ |
| Clean Stream | | | | | $76.27_{\pm 0.16}$ | | |

Table 2: Comparison against state-of-the-art TTA methods under the **mixed** corruption scenario. We report the classification accuracy (%) on CIFAR-10-C following the adaptation process. The best result under each OOD-contaminated stream is marked in bold.

| Stream | Method | LSUN | SVHN | DTD | LSUN + SVHN | LSUN + DTD | DTD + SVHN |
|---|---|---|---|---|---|---|---|
| Mixed | Test | | | | 64.38 | | |
| | BN | $65.44_{\pm 0.90}$ | $59.00_{\pm 1.07}$ | $58.51_{\pm 0.66}$ | $64.34_{\pm 1.10}$ | $64.58_{\pm 0.92}$ | $58.46_{\pm 0.91}$ |
| | TENT | $65.97_{\pm 1.09}$ | $59.61_{\pm 0.82}$ | $57.52_{\pm 0.96}$ | $64.95_{\pm 1.31}$ | $64.71_{\pm 1.47}$ | $58.21_{\pm 1.03}$ |
| | TENT$_f$ | $65.36_{\pm 1.03}$ | $59.85_{\pm 0.89}$ | $57.16_{\pm 0.72}$ | $64.22_{\pm 1.12}$ | $64.83_{\pm 1.70}$ | $57.84_{\pm 1.98}$ |
| | EATA | $67.89_{\pm 1.38}$ | $60.05_{\pm 1.49}$ | $55.02_{\pm 1.71}$ | $64.34_{\pm 1.11}$ | $64.09_{\pm 0.98}$ | $59.31_{\pm 1.30}$ |
| | SAR | $66.09_{\pm 0.88}$ | $58.99_{\pm 1.08}$ | $58.51_{\pm 0.66}$ | $64.34_{\pm 1.10}$ | $64.58_{\pm 1.68}$ | $58.46_{\pm 1.91}$ |
| | R3 | $\mathbf{67.93}_{\pm 0.89}$ | $\mathbf{61.81}_{\pm 1.60}$ | $\mathbf{58.87}_{\pm 0.83}$ | $\mathbf{66.17}_{\pm 0.88}$ | $\mathbf{67.40}_{\pm 1.61}$ | $\mathbf{62.25}_{\pm 0.92}$ |
| Clean Stream | | | | | $63.12_{\pm 0.89}$ | | |

We additionally validate that LSUN is indeed closer to CIFAR-10-C by quantifying the similarity of OOD datasets to CIFAR-10-C in terms of the 2-Wasserstein distance (Givens & Shortt, 1984). The results and analysis, included in Section A8 of Appendix, uphold that the proximity of LSUN to CIFAR-10-C is what makes it benign OOD-contamination.

We now consider the test scenario in which all types of shifted InD data appear together with no boundary. Likewise, OOD data are corrupted with a mixture of corruption types. We compare the classification accuracy at the end of the adaptation process in Table 2. We observe that LSUN again improves the performance of TENT, demonstrating that it can serve as a benign form of OOD-contamination in different test scenarios. Across all OOD-contamination types, R3 consistently attains the best performance among all the compared approaches.

### 5.3 Validation on a Larger Dataset

We now verify that the effectiveness of R3 can be scaled to a more complex dataset through experiments on OOD-contaminated ImageNet-C streams. Analogous to the CIFAR-10-C experiments, we compare all approaches under both separate and mixed scenarios. We present the performance of R3 and those of compared approaches in Table 3. R3 exhibits a significant degree of performance improvement under both deployment scenarios and across various OOD-contamination types. These results provide concrete evidence for the scalability and universality of R3.

### 5.4 Additional Results and Discussion

**(1) Compatibility with Various TTA Algorithms:** R3 is primarily designed for and implemented in conjunction with the entropy minimization-based loss for TTA. To demonstrate that R3 can be utilized with a broader range of TTA algorithms, we combine R3 with TTA methods based on pseudo-labeling (Goyal et al., 2022) and augmentation invariance (Zhang et al., 2022) and report the results in Table A1 of Appendix. The results clearly show that the performance improvement brought upon by R3 is not exclusive to the entropy minimization-based algorithm for TTA.
**(2) Performance on Clean Test Streams:** Through comparison against state-of-the-art TTA approaches in Table A2 of Appendix, we validate that R3 can be used for adaptation on clean test

Table 3: Comparison against state-of-the-art TTA approaches on ImageNet-C under separate and mixed corruption scenarios. The best classification accuracy (%) in each column is marked in bold.

| Method | Separate | | | Mixed | | |
|---|---|---|---|---|---|---|
| | iNat | DTD | iNat + DTD | iNat | DTD | iNat + DTD |
| Test | ———— 26.65 ———— | | | ———— 25.52 ———— | | |
| BN | $25.62_{\pm 0.14}$ | $27.57_{\pm 0.15}$ | $27.83_{\pm 0.11}$ | $16.37_{\pm 0.85}$ | $17.60_{\pm 0.78}$ | $19.21_{\pm 0.10}$ |
| TENT | $26.09_{\pm 0.07}$ | $27.90_{\pm 0.15}$ | $28.57_{\pm 0.21}$ | $15.98_{\pm 0.80}$ | $15.26_{\pm 1.35}$ | $17.89_{\pm 0.57}$ |
| TENT$_f$ | $25.71_{\pm 0.14}$ | $27.57_{\pm 0.14}$ | $27.92_{\pm 0.09}$ | $14.17_{\pm 1.33}$ | $15.69_{\pm 1.67}$ | $17.36_{\pm 0.97}$ |
| EATA | $25.81_{\pm 0.12}$ | $27.47_{\pm 0.15}$ | $27.30_{\pm 0.33}$ | $17.30_{\pm 0.46}$ | $17.28_{\pm 0.71}$ | $18.31_{\pm 0.67}$ |
| SAR | $25.61_{\pm 0.14}$ | $27.57_{\pm 0.15}$ | $27.83_{\pm 0.11}$ | $16.38_{\pm 0.72}$ | $16.93_{\pm 0.66}$ | $19.20_{\pm 0.79}$ |
| R3 | $\mathbf{27.50}_{\pm 0.32}$ | $\mathbf{30.83}_{\pm 0.22}$ | $\mathbf{30.58}_{\pm 0.16}$ | $\mathbf{18.43}_{\pm 0.42}$ | $\mathbf{19.50}_{\pm 0.78}$ | $\mathbf{20.99}_{\pm 0.90}$ |
| Clean Stream | $30.75_{\pm 0.09}$ | | | $18.90_{\pm 0.57}$ | | |

streams, void of OOD-contamination. R3 visibly outperforms competitive baselines on both clean test streams, showcasing its capability to handle a variety of test scenarios.

**(3) OOD Detection Metric:** We investigate R3 from the perspective of OOD detection by analyzing H-score, defined as: $H = (2 * ACC_{\text{InD}} * ACC_{\text{OOD}})/(ACC_{\text{InD}} + ACC_{\text{OOD}})$, where $ACC_{\text{InD}}$ and $ACC_{\text{OOD}}$ refer to the classification accuracy on shifted InD data and OOD detection accuracy, respectively. In Table A3 of Appendix, H-scores of compared approaches measured on one of the CIFAR-10-C + DTD streams are reported. R3 successfully improves the classification accuracy on shifted InD data while maintaining competitive OOD detection accuracy. This result further elucidates that some OOD instances are more beneficial for TTA than others because the compared approaches perform comparably on $ACC_{\text{OOD}}$ but still show disparities in $ACC_{\text{InD}}$. Moreover, while R3 does not lead to a noticeable improvement in $ACC_{\text{OOD}}$, it is capable of identifying these benign OOD instances and incorporating them into the adaptation process to effectively boost $ACC_{\text{InD}}$.

## 6 ABLATION STUDY AND HYPERPARAMETER SENSITIVITY ANALYSIS

### 6.1 COMPONENT-WISE ABLATION STUDY

In Figure 4, we visualize the change in the adaptation performance of R3 on CIFAR-10-C + SVHN and ImageNet-C + DTD streams as the filtering scheme and each one of the loss functions are applied incrementally. Each technical component clearly contributes to improving the performance of R3, allowing it to achieve superior performance as demonstrated in earlier sections.

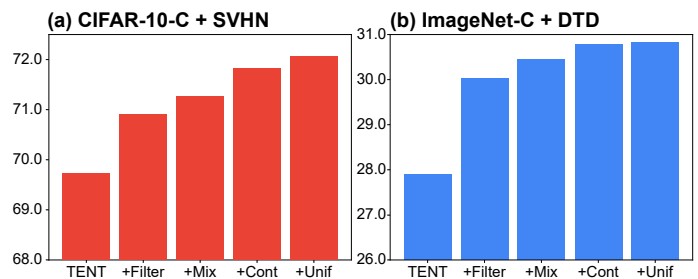

Figure 4: Ablation study on two different OOD-contaminated streams: (a) CIFAR-10-C + SVHN and (b) ImageNet-C + DTD.

Furthermore, we explore different design choices for the filtering scheme and mixup component in R3. First, we study the results of solely using energy-based filtering for compared approaches in Table A4. Two conclusions can be derived from these results. 1) Utilizing energy-based filtering deteriorates the performance of compared approaches in most cases. This result indicates that the use of a more competitive OOD detection method does not always lead to improvement in adaptation performance, confirming the existence of beneficial OOD data. 2) R3 performs better with the proposed dual filtering scheme, indicating that the conservative filtering method in R3 is more effective at conserving beneficial OOD data for TTA than conventional OOD detection scores.

Second, we replace similarity-based mixup in R3 with two more commonly-used forms of mixup - original randomized mixup and CutMix Yun et al. (2019) - and compare their performances in Table A5. Utilizing the proposed similarity-based mixup consistently outperforms the other two, justifying our design of similarity-based mixing coefficients. Collectively, these results support that

Table 4: Results of using different batch sizes for online adaptation. R3 consistently surpasses the two strongest baseline approaches.

| BS | Method | C10-C + SVHN | ImgNet-C + DTD |
|---|---|---|---|
| $\times 2$ | TENT | 70.46 | 30.25 |
| | SAR | 70.44 | 30.01 |
| | R3 | 72.05 | 31.84 |
| $\times \frac{1}{2}$ | TENT | 66.60 | 23.69 |
| | SAR | 68.55 | 23.52 |
| | R3 | 69.52 | 28.72 |

Table 5: Results of using different InD-to-OOD data ratios. R3 comes on top of the two other baselines even if the data ratio is changed.

| Ratio | Method | C10-C + SVHN | ImgNet-C + DTD |
|---|---|---|---|
| $1 : 2$ | TENT | 67.15 | 27.29 |
| | SAR | 66.25 | 27.07 |
| | R3 | 68.92 | 27.68 |
| $2 : 1$ | TENT | 73.48 | 31.72 |
| | SAR | 72.45 | 31.30 |
| | R3 | 74.98 | 33.01 |

although R3 relies on existing ideas, deliberate modifications introduced in R3, *e.g.,* re-defining the mixing coefficient or employing conservative dual filtering, play a critical role in its success.

## 6.2 HYPERPARAMETER SENSITIVITY ANALYSIS

**Data stream configuration** To show that R3 is robust to changes in test scenarios, we report the results of altering two major factors in data stream configuration: the batch size used for online adaptation and the ratio of InD to OOD data. In Table 4, we report the results of doubling and halving default batch sizes used for CIFAR-10-C and ImageNet-C streams. R3 consistently comes on top regardless of the batch size setting. In particular, R3 improves the performance on ImageNet-C + DTD stream by almost 5%p when the batch size is halved. According to Table 5, R3 successfully maintains its competitive performance even when the ratio of InD-to-OOD data changes.

**Filtering thresholds & Loss coefficients** We perform a hyperparameter sensitivity analysis of two filtering thresholds and four weighting coefficients for loss terms to further confirm the stability of R3. Figure 5 (a) visualizes changes in R3 performance according to different threshold values ($\tau_{\mathrm{es,cs}}$). Similarly, changes in the performance of R3 according to different coefficients for the loss terms ($\lambda_{\mathrm{Ent,Mix,Cont,Unif}}$)

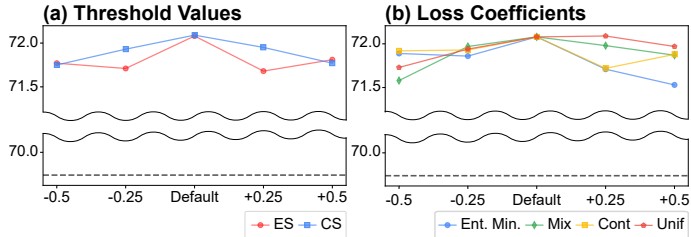

Figure 5: Sensitivity of R3 to (a) different values of the energy score (red) and cosine similarity (blue) thresholds; and (b) to varying coefficients for the four separate loss terms.

are shown in Figure 5 (b). All other hyperparameters remain unchanged. The grey dotted lines indicate the TENT performance. Within the tested range, R3 steadily outperforms TENT, a strong baseline approach, suggesting that finding the optimal set of hyperparameters is not too difficult.

## 7 CONCLUDING REMARKS

This paper studied for the first time TTA with OOD-contaminated data streams, a realistic TTA scenario of grave importance. We unearthed the existence of benign OOD data that can improve, rather than harm, the adaptation performance on shifted InD data. To delve into the intriguing phenomenon of benign OOD-contamination, we empirically analyzed the feature space of the pre-trained DNN; our analysis revealed that two types of data share domain-specific characteristics, allowing some OOD data to aid in the domain transfer process in TTA. Motivated by such analytical results, we proposed R3, a novel TTA algorithm designed for OOD-contaminated streams, and showcased its effectiveness and versatility through extensive experiments that span combinations of two shifted InD datasets and four OOD datasets. As a pioneering investigation of its kind, this work will contribute to promoting the safe and robust deployment of pre-trained DNNs in an open world.

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

## A1    BASELINES AND IMPLEMENTATION DETAILS

Following are the baseline methods from TTA literature that R3 is compared to.

- Test: evaluates the pre-trained DNN on new test data without additional modification.
- BN Adapt: replaces the Batch Norm statistics of the pre-trained DNN with those of test data. We use the abbreviation "BN" to refer to this baseline approach.
- TENT (Wang et al., 2020): updates the statistics and affine parameters of Batch Norm layers by minimizing the entropy-based loss. TENT with filtering (TENT$_f$) performs TENT after removing high-entropy instances. The filtering threshold is set to be $\log(1000) * 0.04$.
- EATA (Niu et al., 2022): utilizes the Fisher regularizer to preserve important parameters and performs instance selection and re-weighting based on the DNN's predictive entropy.
- SAR (Niu et al., 2023): discards unstable test instances with large gradients and replaces the vanilla SGD optimizer with a sharpness-aware minimization optimizer.

All our experiments are implemented using PyTorch (Paszke et al., 2019) and conducted with NVIDIA V100 GPU. We use ResNet-50 (He et al., 2016) with Batch Norm layers for experiments on CIFAR-10-C and ImageNet-C. For CIFAR-10 pre-training, we use the SGD optimizer with the initial learning rate of 0.1, annealed at cosine rate, momentum of 0.9, and weight decay of 0.0005. We use the model provided by the PyTorch timm (Wightman, 2019) as the pre-trained model for ImageNet-C. For the adaptation process, we use the SGD optimizer with a learning rate of 0.0025 and momentum of 0.9. We use the batch size of 32 and 64 for CIFAR-10-C and ImageNet-C, respectively. The default ratio of InD to OOD data is set to be 1:1.

## A2    COMPATIBILITY WITH OTHER TTA FRAMEWORKS

Table A1: Compatibility with a broader range of TTA algorithms. PL refers to the pseudo-labeling-based method. MEMO is a method based on augmentation invariance.

| Stream | Method | CIFAR-10-C + | | | ImageNet-C + | |
| | | LSUN | SVHN | DTD | iNat | DTD |
|---|---|---|---|---|---|---|
| Separate | PL | 75.78 | 70.56 | 70.97 | 26.21 | 28.08 |
| | + R3 | 76.64 | 71.79 | 72.63 | 27.53 | 29.32 |
| | MEMO | 76.51 | 74.02 | 74.54 | 27.86 | 27.47 |
| | + R3 | 76.83 | 74.95 | 75.24 | 28.67 | 28.98 |
| Mixed | PL | 66.10 | 60.27 | 56.62 | 17.11 | 16.52 |
| | + R3 | 67.65 | 61.69 | 58.63 | 18.26 | 17.73 |
| | MEMO | 67.65 | 61.69 | 58.63 | 18.26 | 17.41 |
| | + R3 | 67.32 | 65.36 | 63.85 | 17.89 | 19.04 |

## A3    ADAPTATION RESULTS ON CLEAN STREAMS

Table A2: Adaptation results on "clean" streams without OOD data.

| Stream | Dataset | EATA | SAR | R3 |
|---|---|---|---|---|
| Sep | CIFAR-10-C | 78.19 | 78.01 | 79.66 |
| | ImageNet-C | 30.14 | 30.81 | 31.79 |
| Mix | CIFAR-10-C | 63.36 | 64.57 | 65.01 |
| | ImageNet-C | 17.82 | 19.37 | 19.92 |

## A4  OOD DETECTION PERFORMANCE

Table A3: H-score analysis on one of the CIFAR-10-C + DTD streams. H-score allows us to simultaneously consider the classification accuracy on shifted InD data and OOD detection accuracy. R3 again exceeds other strong baselines in terms of H-score.

|       | $ACC_{\text{InD}}$ | $ACC_{\text{OOD}}$ | H-score |
|-------|-------|-------|---------|
| TENT$_f$ | 69.70 | 53.00 | 60.21 |
| EATA  | 68.40 | 52.80 | 59.59 |
| SAR   | 68.39 | 53.34 | 59.94 |
| R3    | 72.50 | 53.38 | 61.89 |

## A5  ADDITIONAL DESIGN CHOICE EXPLORATION

Table A4: The filtering component in all compared approaches is replaced with OOD filtering based on the energy score, an advanced metric of OOD-ness.

| Stream | Method | CIFAR-10-C + | | | ImageNet-C + | |
|--------|--------|------|------|------|------|------|
|        |        | LSUN | SVHN | DTD | iNat | DTD |
| Sep | TENT | 74.82 | 68.61 | 68.62 | 25.73 | 26.80 |
|     | EATA | 74.74 | 68.62 | 68.24 | 24.66 | 27.17 |
|     | SAR  | 74.91 | 68.57 | 68.58 | 24.73 | 27.21 |
|     | R3   | 76.46 | 70.82 | 70.97 | 26.29 | 29.07 |
| Mix | TENT | 65.85 | 58.99 | 56.51 | 15.70 | 17.23 |
|     | EATA | 65.69 | 60.17 | 55.02 | 17.05 | 18.06 |
|     | SAR  | 66.09 | 58.99 | 57.63 | 17.08 | 18.14 |
|     | R3   | 66.91 | 61.27 | 58.02 | 17.79 | 18.88 |

Table A5: Results of executing R3 with original mixup and CutMix.

| Stream | Method | CIFAR-10-C + | | | ImageNet-C + | |
|--------|--------|------|------|------|------|------|
|        |        | LSUN | SVHN | DTD | iNat | DTD |
| Sep | Original | 76.61 | 71.76 | 71.31 | 26.07 | 27.80 |
|     | CutMix   | 76.61 | 70.93 | 70.88 | 25.43 | 27.39 |
|     | Ours     | 76.64 | 72.08 | 72.41 | 27.50 | 30.83 |
| Mix | Original | 66.74 | 60.10 | 58.09 | 16.98 | 18.02 |
|     | CutMix   | 66.71 | 60.27 | 57.59 | 16.95 | 17.51 |
|     | Ours     | 67.93 | 61.81 | 58.87 | 18.43 | 19.50 |

Table A6: Results of replacing the filtering component in other algorithms with the proposed filtering scheme, which is denoted by R.F., a shorthand for R3 filtering scheme.

| Stream | Method | CIFAR-10-C + | | | ImageNet-C + | |
| | | LSUN | SVHN | DTD | iNat | DTD |
|---|---|---|---|---|---|---|
| Sep | EATA | 76.12 | 69.81 | 68.25 | 25.81 | 27.47 |
| | + R.F. | 76.30 | 71.45 | 71.50 | 26.87 | 28.90 |
| | SAR | 76.17 | 69.45 | 68.53 | 25.61 | 27.57 |
| | + R.F. | 76.26 | 71.08 | 69.45 | 25.97 | 28.91 |
| Mix | EATA | 67.89 | 60.05 | 55.02 | 17.30 | 17.28 |
| | + R.F. | 67.89 | 61.07 | 56.95 | 17.93 | 17.92 |
| | SAR | 66.09 | 58.99 | 58.51 | 16.38 | 16.93 |
| | + R.F. | 66.27 | 59.36 | 58.62 | 17.08 | 17.57 |

## A6    OVERALL ALGORITHM FOR R3

---
**Algorithm 1:** Test-Time Adaptation with R3
---
1 **Require:** Pre-trained DNN $f_\theta(x)$, Class-wise Prototypes $r_c$, Test Data Batch $\mathcal{D}_{\text{te}} = \{(x_{\text{te}}^i)\}_{i=1}^B$
2     Compute $ES(x_{\text{te}}^i; f_\theta)$ and $CS(x_{\text{te}}^i; f_\theta)$ for $x_{\text{te}}^i \in \mathcal{D}_{\text{te}}$
3     Determine $S_\text{B}$ and $S_\text{M}$ according to Eq. (2)
4     Compute $\alpha$ and generate $x_{\text{mix}}$ for $x_\text{B} \in S_\text{B}$ according to Eq. (3)
5     Determine $S_{\text{mix}}$ according to Eq. (2)
6     Compute two entropy minimization losses $\mathcal{L}_{\text{Ent}}(x_\text{B})$ and $\mathcal{L}_{\text{Mix}}(x_{\text{mix}})$, where $x_{\text{mix}} \in S_{\text{mix}}$
7     Compute contrastive loss $\mathcal{L}_{\text{Cont}}$ with $S_{\text{ind}}, S_{\text{mix}}$, and $r_c$ according to Eq. (4)
8     Compute uniform loss $\mathcal{L}_{\text{Unif}}$ with $S_\text{M}$
9     Update the parameters of Batch Normalization layers using $\mathcal{L}_{\text{R3}}$
---

## A7    R3 HYPERPARAMETER CONFIGURATION

In Table A7, we report a detailed hyperparameter configuration used for R3. The same set of hyperparameters is used for each stream under separate and mixed corruption scenarios. After each update step $k$, the filtering thresholds and loss coefficients are adjusted at the rate of $\gamma$ as follows: $\tau_{k+1} = 0.9 * \tau_k + (0.1 * \gamma)\tau_k$, and $\lambda_{k+1} = 0.9 * \lambda_k + (0.1 * \gamma)\lambda_k$.

Table A7: R3 Hyperparameter configurations for different OOD-contaminated streams.

| Stream Type | $\tau_{\text{es}}$ | $\tau_{\text{cs}}$ | $\lambda_{\text{Ent}}$ | $\lambda_{\text{Mix}}$ | $\lambda_{\text{Cont}}$ | $\lambda_{\text{Unif}}$ | $\gamma$ |
|---|---|---|---|---|---|---|---|
| CIFAR-10-C + | | | | | | | |
|    LSUN | 6.0 | 0.8 | 1.8 | 1.8 | 0.1 | 0.1 | 0.999 |
|    SVHN | 6.0 | 0.8 | 2.0 | 2.0 | 1.5 | 1.5 | 0.999 |
|    DTD | 6.0 | 0.8 | 2.0 | 2.0 | 1.5 | 1.5 | 0.999 |
|    LSUN + SVHN | 6.0 | 0.8 | 2.0 | 2.0 | 1.5 | 1.5 | 0.999 |
|    LSUN + DTD | 6.0 | 0.8 | 2.0 | 2.0 | 1.5 | 1.5 | 0.999 |
|    DTD + SVHN | 6.0 | 0.8 | 2.0 | 2.0 | 1.5 | 1.5 | 0.999 |
| ImageNet-C + | | | | | | | |
|    iNat | 0.5 | 0.45 | 3.0 | 3.0 | 3.0 | 3.0 | 0.8 |
|    DTD | 0.5 | 0.45 | 5.0 | 3.5 | 2.0 | 0.5 | None |
|    iNat + DTD | 0.5 | 0.45 | 5.0 | 3.5 | 2.0 | 0.5 | None |

We elaborate on observations regarding the trends in the hyperparameter configuration that provide meaningful insights into how these values can be tuned prior to deployment. First, the two $\tau$ values appear to be closely correlated with the number of classes in the InD dataset; the optimal values of $\tau_{\text{es}}$ and $\tau_{\text{cs}}$ for ImageNet-C streams are smaller than those for CIFAR-10-C streams. $\tau_{\text{es}}$ is based on the model's predictive entropy distribution, and as the number of classes in the InD dataset grows, the model will struggle to output a highly confident prediction on one specific class. Therefore, we would want to use a smaller value for $\tau_{\text{es}}$ as the number of classes increases. $\tau_{\text{cs}}$ utilizes the distance between individual features and class-wise prototypes. With a greater number of class-wise prototypes populating the feature space, the distance between an individual feature and its nearest class-wise prototype is likely to decrease.

Second, for $\lambda$ values, with the exception of streams that exclusively contain near OOD data (e.g., CIFAR-10-C+LSUN and ImageNet-C+iNaturalist), the optimal hyperparameter settings remain mostly consistent across test streams within the same InD dataset. In real deployment scenarios, it is

more likely that some mixture of OOD data types will occur. Therefore, when one cannot explicitly predict the type of OOD data that will be present in test streams in advance, the hyperparameter settings used for the rest of test streams would generally be recommended.

## A8 DATASET ANALYSIS WITH 2-WASSERSTEIN DISTANCE

In this section, we compare how close each OOD dataset is to CIFAR-10-C using 2-Wasserstein distance (Givens & Shortt, 1984) ($W$) as the measure of similarity. $W$ between CIFAR-10-C and LSUN, SVHN, and DTD are 2.48, 3.62, and 3.53, respectively. We demonstrated in the main paper that LSUN, which is closest to CIFAR-10-C, functions as a benign type of OOD-contamination, while SVHN and DTD, which are relatively farther away, have detrimental effects on the adaptation performance. Therefore, this quantitative analysis further that benign OOD-contamination is induced by the closeness of OOD data to shifted InD data.

