# OpenReview forum: "Reduce, Reuse, and Recycle: Navigating Test-Time Adaptation with OOD-Contaminated Streams"
_ICLR.cc/2024/Conference — Submitted to ICLR 2024_

### Official Review · Reviewer_8qGX · 2023-10-30

**Soundness:** 3 good
**Presentation:** 3 good
**Contribution:** 2 fair
**Rating:** 5
**Confidence:** 3

**Summary:**

This paper aims to tackle test-time adaptation in an open-world setting where data from unseen classes (OOD data) and seen classes (InD data) are mixed up in the test data stream under the same distribution shift. Based on empirical observation, this paper claims that the existence of OOD data can benefit the adaptation of the pre-trained model on shifted InD data. A TTA algorithm is further proposed with OOD data filtering and recycling, similarity-based data mixup, and contrastive learning with class-wise prototypes. Experimental results on CIFAR-10-C and ImageNet-C are illustrated. No theory is provided.

**Strengths:**

1. This paper considers test-time adaption in the open world, which is interesting and realistic.

2. This paper tackles the OOD-contaminated distribution-shifted data streams by rigorously filtering out OOD data and then learning the shifted InD and OOD data with different training objectives.

3. Experimental results on CIFAR-10-C and ImageNet-C show that the proposed algorithm outperforms the baselines under single and mixed corruption scenarios.

**Weaknesses:**

1. The claim of "benign OOD contamination" is not strong enough:
   - Experimental settings are lacking to show a fair comparison between test streams with and without OOD contamination.
   - The total number of data samples is not mentioned in the comparison. If test streams with OOD data have more data than those without OOD data, there is not enough evidence to show that the improvements come from auxiliary signals provided by OOD data rather than the increase of shifted data samples.
   - Only batch norm adaptation is discussed and analyzed.

2. The discussion about benign and harmful OOD data is not sufficient.
   - According to which measure can we distinguish benign OOD data and harmful OOD data?
   - Does the advancement of R3 come from the removal of OOD data or the preservation of benign OOD data? The main experiment results show that R3 only outperforms the clean stream baseline under a few OOD-contaminated scenarios. Does it mean that benign OOD data is also removed during the filtering process?

3. The reason for using cosine similarity is not clearly stated. Using cosine similarity alone as the filtering scheme is also recommended in the ablation study.

4. Why perform the filtering process on mixed shifted InD data?

5. Discussions regarding the effect of batch size and the ratio of InD to OOD data on the performance of R3 are also recommended.

6. Although it outperforms baselines in the mixed corruption scenario, R3 still fails to improve the pre-trained model on CIFAR-10-C+SVHN/DTD/DTD+SVHN and ImageNet-C with OOD data.

**Questions:**

Please [Weaknesses]

---

> ### Author Response · Authors · 2023-11-22
> **Author Response**
>
> We would like to first thank Reviewer 8qGX for the thorough feedback. We hope our response below addresses the concerns and questions raised by the reviewer.
>
> W1. Claim of benign OOD contamination
> - In our work, we provide two pieces of evidence for benign OOD-contamination.  First, we reveal that in certain test streams, OOD data can improve the performance of TENT, which does not employ any filtering scheme. Second, we demonstrate that algorithms with similar OOD detection performance can exhibit wildly different InD adaptation performance, indicating that OOD data have different degrees of helpfulness.
> - By default, a test stream contains the same number of InD and OOD samples.
> - Because updating the learnable parameters during TTA will likely result in model collapse, it is commonly assumed that only batch norm adaptation is performed at test-time.
>
> W2. Discussion about harmful OOD data
> - Generally speaking, if OOD data help adaptation on shifted InD performance, we refer to it as benign OOD data, and vice versa. Therefore, while it is difficult to determine the harmfulness of OOD data without post-adaptation performance, we speculate that OOD data that exist close to InD data in the feature space of the pre-trained model will tend to be benign. This hypothesis is supported through analyses of t-sne plots and 2-Wasserstein distance measurements.
>
> - The performance improvement of R3, compared to other TTA baselines, can be attributed to 1) its increased distinguishability between benign vs. harmful OOD instances and 2) effective utilization of benign OOD instances. R3 only outperforms the clean stream because while benign OOD data exist, harmful OOD data are still quite dominant, and R3, as well as all of the existing baselines, are incapable of perfectly picking out shifted InD and benign OOD data for adaptation. Yet, the significant relative performance improvement of R3 from TTA baselines suggests that R3 does indeed extend the state-of-the-art in this particular TTA scenario.
>
> W3. The cosine similarity is used as an auxiliary metric for filtering OOD instances because we believe that those lie closer to the InD data in the feature space aid better in the domain transfer process.
>
> W4. Without access to explicit test labels, it is impossible to distinguish between the shifted InD and OOD data. Therefore, R3 is bound to be blind to this distinction, leaving it with no choice but to perform a filtering process on both types of data.
>
> W5. The results of ablation studies on the batch size and ratio of InD to OOD data are reported in Tables 4 and 5 of the main paper, respectively.
>
> W6. We were not exactly sure what the reviewer was referring to here by the “CIFAR-10-C+SVHN/DTD/DTD+SVHN and ImageNet-C with OOD data.” If the mentioned quantity refers to the “Clean Stream” accuracy in Tables 1, 2, 3, and 4, it was obtained by performing TENT on clean streams to provide the potential empirical upper bound not by performing inference with a pre-trained model.

---

> > ### Comment · Reviewer_8qGX · 2023-11-23
> > **Re: response**
> >
> > Thanks for your response. However, the concerns about motivations and experimental settings are still unclear. Thus, I will keep my original score the same, but I would like to note that I will not fight for rejection, as it is borderline.

---

### Official Review · Reviewer_szCP · 2023-11-01

**Soundness:** 2 fair
**Presentation:** 3 good
**Contribution:** 2 fair
**Rating:** 3
**Confidence:** 4

**Summary:**

This paper studies the problem of TTA with OOD-contaminated data streams. They empirically reveal that the existence of benign OOD data can improve the adaptation performance on shifted ID data. Then, they introduce a stage-wise approach, term R3, to aid in the domain transfer process. Experimental results show that R3 improves baseline methods on OOD-contaminated streams created with CIFAR-10-C and ImageNet-C.

**Strengths:**

- The paper is well organized and with clear clarification.

- Based on the experiment results, this paper provides some findings or analyses.

**Weaknesses:**

- My main concern is the motivation. Using external OOD data during training has empirically proved to be effective in OOD detection/generalization [r1]. However, in this manuscript, the authors appear to replicate these observed phenomena without introducing novel insights or furnishing theoretical guarantees specific to test-time adaptation. This redundancy raises questions about the contribution and originality of the work, necessitating a more comprehensive justification to substantiate the research's value. Moreover, the term 'benign' in the manuscript is vaguely defined, leading to potential confusion about what constitutes 'harmful' or 'malignant' in this context. Clarification from the authors would enhance the paper's clarity.

- The use of t-SNE visualization for validating assumptions on benign OOD data in the manuscript is notably heuristic and may not be entirely reliable. This situation gives the impression that the authors are attempting to validate one intuitive hypothesis using another intuitive tool, which significantly weakens the theoretical foundation of the paper.

- The proposed method simply concatenates three stages without offering additional insights, particularly concerning the so-called `benign OOD data'. The authors need to delve deeper and elucidate how these stages interact and contribute to handling benign OOD data.

- The experimental results indicate that the proposed method only achieves marginal improvements, which raises further doubts about whether the motivation of the paper has been accurately conveyed and validated. The authors need to address this issue by providing a more thorough analysis of the results and discussing potential reasons for the limited improvement, ensuring that the paper's objectives and contributions are clearly and convincingly presented.

[r1] Generalized Out-of-Distribution Detection: A Survey.

**Questions:**

Please refer to the weaknesses.

---

> ### Author Response · Authors · 2023-11-22
> **Author Response**
>
> We first thank Reviewer szCP for the constructive critique of our paper. With our additional discussion below, we sincerely hope that the reviewer will reconsider the research value of our work.
>
> The term “benign OOD data” in the manuscript refers to OOD data that can improve the adaptation performance on the shifted InD data; any OOD data that does not meet this criterion are considered to be harmful or malignant.
>
> We agree that OOD data have been demonstrated to improve the OOD detection or generalization performance, but this is the first work to study its effect on test-time adaptation. Because intuition that holds in one domain may not always extend to another, it would be a stretch to regard the observed phenomenon of benign OOD-contamination in TTA as simple regurgitation of previously-known facts.
>
> We are deeply disheartened that the reviewer finds the insights from our empirical analyses lacking in research value. While our work may not offer the theoretical depth that the reviewer expects, we kindly ask Reviewer szCP to reconsider the value of our work based on the following grounds:
>
> 1. As noted above, we would like to emphasize that this is the first time OOD-contaminated streams have been explored in test-time adaptation. We believe this is a realistic and important setting that significantly broadens the scope and versatility of TTA.
> 2. We do make a meaningful attempt to explicate the reason behind the existence of benign OOD data through analyses of t-sne plots and 2-Wasserstein distance measurements.
> 3. We develop a novel algorithm which is grounded on the empirical findings, as well as extending the state-of-the-art across a variety of test scenarios that span combinations of two shifted InD datasets and five OOD datasets. R3 employs conservative dual filtering to preserve benign data, and new R3 loss functions are designed to amplify the effect of OOD data during the adaptation process through mixup and contrastive learning. Moreover, R3 may only lead to marginal performance improvements in certain test scenarios, we would like to bring to the reviewer’s attention the consistent improvement observed across the board.

---

### Official Review · Reviewer_hsXp · 2023-11-03

**Soundness:** 3 good
**Presentation:** 2 fair
**Contribution:** 2 fair
**Rating:** 5
**Confidence:** 5

**Summary:**

This paper focuses on TTA with data streams contaminated with out-of-distribution (OOD) data. The authors analysis the reasons for of benign OOD-contamination. And they propose a R3 (Reduce, Reuse, and Recycle) TTA algorithm tailored to OOD-contaminated streams.

**Strengths:**

1.	The authors propose metrics of energy and similarity to identify harmful OOD instances.
2.	The authors propose a sample mixup method to alleviate the noisy learning signal from test labels.
3.	The proposed method achieves promising performance in the experimental results.

**Weaknesses:**

1.	I am confused about what is “shifted InD data”. In my understanding, “shifted + InD data” = OOD data. However, in the paper, they are different. And the concept “shifted InD data” is first introduced in the last line of page 1 with further definition. Could the authors give more clear explanation?
2.	In Section 3.1, the concept “benign OOD-contamination” means the OOD data that is helpful for TTA?
3.	In Section 4.1, for “Reduce” part, what are the differences from and advantages over EATA? Can we identify harmful OOD instances with entropy metric instead of energy one?
4.	In “Preliminaries and Notations”, the authors mention that irrelevant OOD data do not share the label set with training data? How to deal with these differences in label sets?
5.	The pipeline seems to be very complex and computationally costly. It may introduce significant latency while inferring models. Do the authors compare the latency of the proposed methods and SOTAs? Note that introducing too much computational cost will make TTA algorithms hard for practical applications.

**Questions:**

Some key concepts are not described clearly in the current manuscript. And some motivation behind current technique solutions remains unclear. If the authors can address my concerns, I would raise my scoring.

---

> ### Author Response · Authors · 2023-11-22
> **Author Response**
>
> We would like to thank Reviewer hsXP for the constructive comments. In the author response below, we provide answers to the questions raised by the reviewer. We sincerely hope that our answers will serve to clarify any confusion.
>
> W1. Distinction between Shifted InD and OOD data
> In the proposed scenario, we assume that two types of out-of-distribution data exist in the test stream: those that share the label space with the in-distribution data but have undergone some type of corruption in the input space (shifted InD data) and those that exist in a completely different label space (OOD data). The objective of test-time adaptation is to adapt the pre-trained model on the shifted InD data, and thus, we consider the latter “OOD-contamination.” As the reviewer correctly notes, “shifted InD data” are commonly considered to be “OOD data,” and we apologize that our wording led to confusion.
>
> W2. Yes, the concept of “benign OOD-contamination” refers to OOD data that could assist in the domain transfer from InD to shifted InD data and consequently lead to the adaptation accuracy on the shifted InD data.
>
> W3. In the reduce part, we utilize both the energy score and the similarity measure to identify and discard OOD instances. In Table A4, we compare the InD adaptation and OOD detection performance of EATA vs. R3. While the two yield a similar detection performance, R3 noticeably outperforms EATA, demonstrating that the filtering schemes in EATA and R3 may appear similar on the surface, the latter is more effective for the proposed scenario.
>
> W4. Yes, the irrelevant OOD data do not share the label space with training data. Because we do not have access to the test labels, when performing adaptation, R3 remains blind to the distinction between the two and applies the filtering scheme to both relevant shifted InD and irrelevant OOD data.
>
> W5. While R3 does lead to latency increase compared to TENT, we would like to argue that such an increase in the computational cost does not completely undermine the practicality of our method, especially because this is the first time OOD-contamination has been studied in TTA. Furthermore, many of previous TTA methods have utilized techniques that can increase the computational cost, and optimizing the computational cost of TTA is a separate research topic on its own.

---

> ### Comment · Reviewer_hsXp · 2023-11-23
> **Responses to Authors**
>
> Thank you for the authors' responses.
>
> Upon careful examination of the rebuttal and the revised manuscript, I have decided to maintain my initial scoring since I still have some concerns:
>
> - Although the authors have clarified the concept of "shifted InD data," I was unable to identify the corresponding modification in the revised manuscript. Additionally, I recommend a rewrite of the introduction section to enhance the clarity of key concepts, as the current version is challenging to comprehend.
>
> - Regarding the distinctions from and advantages over EATA, I encourage the authors to delve into methodological discussions and analyses rather than merely emphasizing empirical results. A more in-depth exploration from a methodological perspective would enhance the overall understanding.
>
> - It appears that the concept of "OOD-contamination" is not original to this work, as EATA has previously highlighted the detrimental effects of high-entropy samples and the potential benefits of relatively low-entropy samples for TTA. I suggest revisiting the novelty claim and acknowledging the existing contributions in this area.
>
> I appreciate the authors' efforts in addressing these concerns and look forward to the continued refinement of the manuscript.

---

### Official Review · Reviewer_hKzM · 2023-11-03

**Soundness:** 3 good
**Presentation:** 2 fair
**Contribution:** 3 good
**Rating:** 5
**Confidence:** 4

**Summary:**

This paper studies an interesting test-time adaptation (TTA) problem, namely TTA with OOD-Contaminated Streams, in which the data stream faces not only sample distribution shifts but also the presence of novel classes. The authors find the existence of benign OOD contamination and conduct experiments to analyze the underlying reasons. Based on the analysis, the authors further propose a R3 (Reduce, Reuse and Recycle) method. Experiments demonstrate the promise of R3. Frankly speaking, I was one of the reviewers when this work was submitted to NeurIPS. For the ICLR version, most of my concerns have been addressed and I only have the following questions and suggestions.

**Strengths:**

The studied problem of TTA with OOD-Contaminated data stream is practical and novel.

Analyses regarding benign OOD-contamination are interesting and contribute new insights.

Experiments demonstrate the effectiveness of the proposed method.

**Weaknesses:**

According to Figure 4, the performance gain is mainly attributed to the "Filter" component, whereas the performance gains from the other components ("Cont" and "Unif") are a bit marginal.

**Questions:**

- Why is there a difference in accuracy between "separate" and "mixed" for "Test (no adapt)" in Tables 1-3? Please include corresponding implementation details in the main paper. Moreover, I highly suggest the authors mix all 15 corrupted datasets rather than sampling a part of the test samples from each corrupted dataset, as the former one is more challenging to demonstrate the effectiveness of the proposed R3 method.

- Could the authors provide information on the number of OOD samples (with novel classes) that are filtered out by your method? This would help further validate the claim of "Benign OOD contamination."

- Will the proposed method still be effective under small batch sizes, e.g., 1, 2, 4? It would be better to provide these results even though it does not perform well enough.

- It would enhance comprehension if clear definitions of "shifted InD" and "OOD" were provided in the introduction. Additionally, it may be more appropriate to utilize alternative terms to describe "shifted InD" and "OOD," as "OOD" inherently encompasses "shifted InD" for the sake of general understanding.

- Many references have been published. Pls carefully check this and cite the corresponding published version.

- It would be better to clearly point out that in Tables 1-3 the clean stream accuracy is measured by Tent. Additionally, in Tables 1-3, the performance gain of Clean Stream (Tent) over “Test (no adapt)” is lower than other TTA papers. I guess this is because the authors use ResNet-50-BN from timm. Here I suggest the authors could also provide comparisons based on ResNet50BN from torchvision and include the results in the Appendix.

——Post Rebuttal——

I thank the authors for their response. However, most of my concerns are still being unaddressed now. The authors did no revisions regarding the Mixed Setting, ShiftID and OOD definition, Results under Small Batch Sizes, Citation Formats, etc. I shall lower my score as I think the current version is still below the high bar of ICLR.

---

> ### Author Response · Authors · 2023-11-22
> **Author Response**
>
> We would like to thank hkzM for the honest and thoughtful review. In response to the reviewer's comments, we provide additional discussion below.
>
> W1. Additional ablation study to demonstrate the effectiveness of R3:
> In Table A6 of the revised paper, we report the results of replacing the filtering scheme in EATA and SAR with the proposed “Filter component.” Even with the proposed filtering scheme, abbreviated as R.F., these approaches fall short of R3, indicating that the additional technical components in R3 are integral parts in the success of R3.
>
> Q1. Mixed Setting Implementation Detail
> As noted in our NeurIPS response, the number of instances in the test set used for the mixed scenario was set to be the same as the number of instances in a single test of shifted InD data in the separate scenario. For instance, for CIFAR-10-c, each shifted test set contains 10,000 samples. To construct the mixed test set for CIFAR-10-c, we sampled 10,000 / 15 number of samples from each test set and shuffled them. Therefore, the test set used for the mixed scenario is different from that used for the separate scenario.
>
> We will incorporate your suggestion to mix all 15 corrupted datasets and include the results in the final version of the paper.
>
> Q2. Number of OOD samples filtered out
> In Table A4 of Appendix, we compare the OOD detection accuracy on a CIFAR-01-C+DTD stream. The total number of OOD samples in this stream are 5000, so the OOD detection accuracy of 53.38% indicates that 2669 samples are being identified as OOD.
>
> While the three compared approaches perform similarly from the perspective of OOD detection, they yield significantly different results on the InD classification task, further justifying that some OOD data are better preserved than discarded.
>
> Q3. Small batch size
> We found that running our method with extremely small batch sizes is rather tricky and leads to complete model collapse mostly due to the filtering factor. Oftentimes, the filtering component would leave us with no sample to perform test-time adaptation with.
>
> Q4. Published references
> We apologize again for not updating the publication statuses of cited references.
>
> Q5. Use of timm model
> We are deeply sorry that this implementation detail was missing; we will include this detail in the final version.

---

### Meta-Review · Area_Chair_qcRV · 2023-12-15

**Metareview:**

Reduce, Reuse, Recycle (R3) addresses an "open-world" extension of the test-time adaptation setting in which test time data may include data with different labels than training time (which this submission calls "OOD-contaminated" data). This work describes the open-world/OOD-contaminated setting, defines an evaluation protocol for it, and proposes the R3 method to filter—reducing and recycling—OOD data and to update by contrastive learning—reusing—to maintain accuracy on unshifted data and improve accuracy on shifted data with both separate or mixed corruptions. Experiments evaluate common test-time adaptation benchmarks (CIFAR-10-C and ImageNet-C) augmented with OOD data from other recognition datasets (LSUN DTD, iNat, SVHN) and R3 achieves equal or better results than standard entropy minimization methods (Tent, EATA, and the more recent SAR).

The argument for the submission is the value of the open-world setting as an extension toward practicality and the variable but positive gains in the open or "contaminated" setting. The argument against the submission is insufficient exposition and experimentation: the exposition was not sufficiently clear (in the definition and naming of InD/OOD/etc. hKzM, hsXp; ) and experiments were not sufficiently comprehensive (hKzM) and significant in the degree of improvement (szCP). While the author-reviewer discussion led to some clarifications, reviewers were not satisfied by the revisions and indicated this by comments and the lowering of their score (hKzM).

The AC sides with rejection but advises the authors to incorporate the feedback from reviewers and resubmit, because the open-world setting is of interest, although in this round the exposition and experiments did not sufficiently support the contributions.

While concurrent work was not considered in the decision for this work, it may nevertheless be helpful for the authors to consider the following papers in a resubmission:

- ODS: Test-Time Adaptation in the Presence of Open-World Data Shift. ICML'23.
- On the Robustness of Open-World Test-Time Training: Self-Training with Dynamic Prototype Expansion. ICCV'23.
- SoTTA: Robust Test-Time Adaptation on Noisy Data Streams. NeurIPS'23.

**Justification For Why Not Higher Score:**

The AC sides with rejection, in agreement with the four expert reviewers (three of whom have specific expertise in test-time adaptation), due to the issues with the exposition and the limitations of the experiments. For the exposition, choice of terminology for "InD" and "OOD" is at odds with the common definition of "OOD" and does not aid comprehension. Furthermore the reviewers have identified more points for clarification (such as the balance of clean/contaminated data during evaluation, whether reference results for closed-world/clean streams are Tent results or source model results, the correct edition of cited works). For the experiments,

Note that certain points were tempered or discounted in this decision: while EATA filtered data it did not consider open-world data (counter to  hsXp), outlier/OOD exposure during training is certainly different than OOD data during testing (counter to szCP), and some requirements of the setting such as blindness to InD/OOD data (counter to W4. by 8qGX). Nevertheless, the balance of the feedback for this round is negative, and after close inspection of the paper the AC does not find a basis to overrule the reviewers.

**Justification For Why Not Lower Score:**

N/A

---

### Decision · Program_Chairs · 2024-01-16

Reject